

# Insight into the maintenance of odontogenic potential in mouse dental mesenchymal cells based on transcriptomic analysis

Yunfei Zheng[1,2,*], Lingfei Jia[1,2,3,*], Pengfei Liu[4], Dandan Yang[4,5], Waner Hu[1], Shubin Chen[4], Yuming Zhao[6], Jinglei Cai[4], Duanqing Pei[4], Lihong Ge[6] and Shicheng Wei[1,3]

[1] Department of Oral and Maxillofacial Surgery, Laboratory of Interdisciplinary Studies, Peking University School and Hospital of Stomatology, Beijing, China
[2] Department of Orthodontics, Peking University School and Hospital of Stomatology, Beijing, China
[3] Central Laboratory, Peking University School and Hospital of Stomatology, Beijing, China
[4] Institute for Stem Cell Biology and Regenerative Medicine, Guangzhou Institute of Biomedicine and Health, Chinese Academy of Sciences, Guangzhou, China
[5] Experimental Center of Pathogenobiology Immunology, Cytobiology and Genetic, College of Basic Medical Sciences of Jilin University, Jilin, China
[6] Department of Pediatric Dentistry, Peking University School and Hospital of Stomatology, Beijing, China
[*] These authors contributed equally to this work.

Corresponding authors
Lihong Ge, gelh0919@126.com
Shicheng Wei, sc-wei@pku.edu.cn

## ABSTRACT

**Background.** Mouse dental mesenchymal cells (mDMCs) from tooth germs of cap or later stages are frequently used in the context of developmental biology or whole-tooth regeneration due to their odontogenic potential. *In vitro*-expanded mDMCs serve as an alternative cell source considering the difficulty in obtaining primary mDMCs; however, cultured mDMCs fail to support tooth development as a result of functional failures of specific genes or pathways. The goal of this study was to identify the genes that maintain the odontogenic potential of mDMCs in culture.

**Methods.** We examined the odontogenic potential of freshly isolated versus cultured mDMCs from the lower first molars of embryonic day 14.5 mice. The transcriptome of mDMCs was detected using RNA sequencing and the data were validated by qRT-PCR. Differential expression analysis and pathway analysis were conducted to identify the genes that contribute to the loss of odontogenic potential.

**Results.** Cultured mDMCs failed to develop into well-structured tooth when they were recombined with dental epithelium. Compared with freshly isolated mDMCs, we found that 1,004 genes were upregulated and 948 were downregulated in cultured mDMCs. The differentially expressed genes were clustered in the biological processes and signaling pathways associated with tooth development. Following *in vitro* culture, genes encoding a wide array of components of MAPK, TGF-$\beta$/BMP, and Wnt pathways were significantly downregulated. Moreover, the activities of *Bdnf*, *Vegfα*, *Bmp2*, and *Bmp7* were significantly inhibited in cultured mDMCs. Supplementation of VEGFα, BMP2, and BMP7 restored the expression of a subset of downregulated genes and induced mDMCs to form dentin-like structures *in vivo*.

**Conclusions.** *Vegfα*, *Bmp2*, and *Bmp7* play a role in the maintenance of odontogenic potential in mDMCs.

## INTRODUCTION

In the field of tooth regeneration, the main concept is to mimic the natural tooth development process using stem cells. Cells or tissues with odontogenic potential are required to regenerate a whole tooth (*Kollar & Baird, 1970b*; *Mina & Kollar, 1987*). Since the odontogenic potential to instruct tooth organogenesis shifts to the dental mesenchyme at bud stage of odontogenesis (*Kollar & Baird, 1970a*; *Mina & Kollar, 1987*), and dental mesenchyme determines the tooth type and size (*Kollar & Baird, 1969*), it will be preferable to use a mesenchymal cell population with odontogenic potential to provide the inductive signal. Nevertheless, the adult dental stem/progenitor cells appear to lack the potency to regenerate an anatomically correct complete tooth organ (*Hu et al., 2014*; *Mao & Prockop, 2012*), and the odontogenic potential of progenitor cells derived from the embryonic stem cells or induced pluripotent stem cells was unknown (*Otsu et al., 2012*; *Ozeki et al., 2013*; *Seki et al., 2015*). Mouse dental mesenchymal cells (mDMCs) isolated from dental mesenchyme of cap or later stages possess the odontogenic potential and they have frequently been used in the context of developmental biology or whole-tooth regeneration (*Honda et al., 2007*; *Ikeda et al., 2009*; *Nakao et al., 2007*). mDMCs are able to induce tooth formation in a dental or nondental epithelial cell population (*Kollar & Baird, 1969*; *Kollar & Baird, 1970b*; *Wang et al., 2010*; *Angelova Volponi, Kawasaki & Sharpe, 2013*; *Cai et al., 2013*). As the preparation of mDMCs is time-consuming and embryos at the right stage are not easily available, *in vitro*-expanded mDMCs could be an available cell source. However, mDMCs fail to support tooth development when they are grown as a monolayer (*Keller et al., 2011*), and the genes that contribute to the loss of odontogenic potential in mDMCs were unknown.

RNA sequencing (RNA-seq) technology provides sensitive detection of global gene expression profiles in cells and tissues, and bioinformatics approaches also examine changes in the context of the overall pathway rather than individual genes (*Wang, Gerstein & Snyder, 2009*). To identify the genes that maintain the odontogenic potential in mDMCs, we evaluated the odontogenic potential of freshly isolated versus cultured mDMCs from the lower first molars in embryonic day 14.5 (E14.5) mice, and compared the transcriptome of freshly isolated mDMCs with that of cultured mDMCs. We found a loss of odontogenic potential accompanied by a significant disturbed transcriptome in cultured mDMCs and identified potential genes that contribute to the loss of odontogenic potential, providing new insight into the maintenance of odontogenic potential of mDMCs.

## MATERIALS & METHODS

### Cell culture

All procedures involving animals described in the present study were reviewed and approved by the Animal Care and Use Committee of Guangzhou Institutes of Health and the Peking

**Table 1  Primers for quantitative real-time PCR.**

|  | Forward | Reverse |
|---|---|---|
| Actin | 5′-GGC TGT ATT CCC CTC CAT CG-3′ | 5′-CCA GTT GGT AAC AAT GCC ATG T-3′ |
| Lhx6 | 5′-CAT TGA GAG TCA GGT ACA GTG C-3′ | 5′-GGG CCG TCC AAA TCA GCT T-3′ |
| Bmp4 | 5′-TTC CTG GTA ACC GAA TGC TGA-3′ | 5′-CCT GAA TCT CGG CGA CTT TTT-3′ |
| Msx1 | 5′- GCA CAA GAC CAA CCG CAAG-3′ | 5′- CGC TCG GCA ATA GAC AGG T-3′ |
| Pax9 | 5′- CAT TCG GCT TCG CAT CGT G -3′ | 5′- CTC CCG GCA AAA TCG AAC C-3′ |
| Fgf10 | 5′- GCA GGC AAA TGT ATG TGG CAT-3′ | 5′- ATG TTT GGA TCG TCA TGG GGA-3′ |
| Bmp2 | 5′-GGG ACC CGC TGT CTT CTA GT-3′ | 5′-TCA ACT CAA ATT CGC TGA GGA C-3′ |
| Fgf3 | 5′-TGC GCT ACC AAG TAC CAC C-3′ | 5′-CAC TTC CAC CGC AGT AAT CTC-3′ |

University in China (permit Number: CMU-B20100106). The lower first molar tooth germs were dissected from the mandibles of E14.5 mice with fine needles and treated with 0.75 mg/ml dispase (Becton, Dickinson and Co., Franklin Lakes, NJ, USA) for 40 min at 37 °C to facilitate separation from dental epithelium. The mesenchymal tissues were then incubated in 0.25% trypsin (Sigma-Aldrich, St. Louis, MO, USA) at 37 °C for 1 min and dissociated with gentle pipetting. The cells were then filtered through a 40 μm cell sieve, resuspended, and cultured in Dulbecco's modified Eagle's medium (DMEM; Gibco, Grand Island, NY, USA) plus 10% fetal bovine serum (FBS; Gibco) (*Duailibi et al., 2004*; *Jiang et al., 2014*; *Yamazaki et al., 2007*; *Zhao et al., 2014*). To maintain the odontogenic potential of mDMCs, supplementation of VEGFα (Sino Biological Inc, Beijing, China), BMP2 (Sigma-Aldrich), and BMP7 (Sigma-Aldrich) were added into the medium.

## Quantitative reverse transcription PCR (qRT-PCR)

Total RNA was extracted with TRIzol (Invitrogen, Carlsbad, CA, USA) and reverse transcribed using an RT-PCR kit (TaKaRa Bio, Otsu, Japan) (*Zheng et al., 2014*). Quantitative PCR was performed in a Thermal CyclerDice7$^{TM}$ Real Time System with SYBR Green Premix EXTaq$^{TM}$ (TaKaRa Bio). The primers used are listed in Table 1. The relative expression of genes was calculated using the $2^{-\Delta Ct}$ method.

## Tissue recombination and subrenal culture

The mDMCs were harvested at indicated time points using 0.25% trypsin (Sigma) and spun down to make cell pellets. The cell pellets were cultured at 37 °C for 2–3 h and then recombined with freshly isolated E14.5 dental epithelium. The recombinants were further cultured *in vitro* for 24 h prior to subrenal culture in adult ICR male mice. The host mice were sacrificed 3 weeks later to harvest the grafted tissue. Grafts were then fixed and subjected to H&E staining for histological analysis.

## RNA isolation and sequencing

To determine the transcriptional regulation following *in vitro* culture, mDMCs from the developing molars in E14.5 mouse embryos were isolated and designated as P0. The cells were then subcultured in standard medium and passaged once they reached 90% confluence, with the first-passage culture designated as P1 and the second-passage culture as P2. P0, P1 and P2 cells were harvested using 0.25% trypsin (Sigma-Aldrich). Total

RNA was extracted using the RNeasy Mini Kit and RNase-Free DNase Set according to the manufacturer's protocol (Qiagen GmbH, Hilden, Germany). The purity and quantity of RNA were assessed using a spectrophotometer (model 8453; Agilent, Santa Clara, CA, USA). RNA libraries for samples were prepared according to instructions for the Illumina TruSeq™ RNA Sample Prep Kit. Sequencing was performed on an Illumina Hiseq™ 2000 (Illumina, San Diego, CA, USA) in duplicate.

## Bioinformatics analysis

Sequenced reads were mapped to the mouse transcriptome (mm10, Ensembl v73) and then aligned using bowtie (v1.0.1) and RSEM (v1.2.12), as described previously (*Hutchins, Takahashi & Miranda-Saavedra, 2015*; *Li & Dewey, 2011*). EDASeq (v1.11.0) was used for GC normalization of samples, and differential expression was called using DESeq2 (v1.12.0). The fold change cut-off was set at twofold and $p$-value < 0.05 were considered to be statistically significant. The Database for Annotation, Visualization and Integrated Discovery (DAVID) was used to determine overrepresented GO categories and KEGG pathways using the entire mouse transcriptome as the background gene set. Protein-protein interaction network of DEGs was constructed using the STRING database (http://string-db.org/). Ingenuity pathway analysis (Ingenuity Systems, Redwood City, CA, USA) was used to analyze the upstream regulators of the differential expression between the samples (*Razali et al., 2015*).

## Statistical analysis

Statistical analyses were performed using SPSS PASW version 18. Experiments with cell cultures were performed at least in triplicate and data are expressed as mean ± standard deviation (SD). The differences among groups were analyzed using one-way ANOVA. A two-tailed $p$-value < 0.05 was considered to be statistically significant.

# RESULTS

mDMCs were isolated from the developing molars in E14.5 mouse embryos and were designated as P0. One part of the P0 cells was recombined with E14.5 dental epithelium and cultured in kidney; another part was subjected to RNA-seq. The remaining cells were subcultured in standard medium, with the first-passage culture designated as P1 and the second-passage as P2. P1 and P2 cells were also split into two portions to ensure an identical cell state for the recombination assay and the RNA-seq (Fig. 1).

## Odontogenic potential of cultured mouse dental mesenchymal cells

mDMCs are typically cultured in Eagle's minimum essential medium plus fetal bovine serum and antibiotics (*Jiang et al., 2014*; *Keller et al., 2011*; *Zhao et al., 2014*). mDMCs showed an atypical round- or spindle-shaped fibroblast-like morphology with a higher nuclear-to-cytoplasmic ratio, indicating their primitive character. Although the cells retained fibroblastic features up to the second passage, they exhibited a senescent phenotype with increased cytoplasm and augmented volume (Fig. 2A).

The odontogenic potential of P0, P1, and P2 cells were examined to identify the impact of the culturing. Although freshly isolated mDMCs developed into well-structured teeth

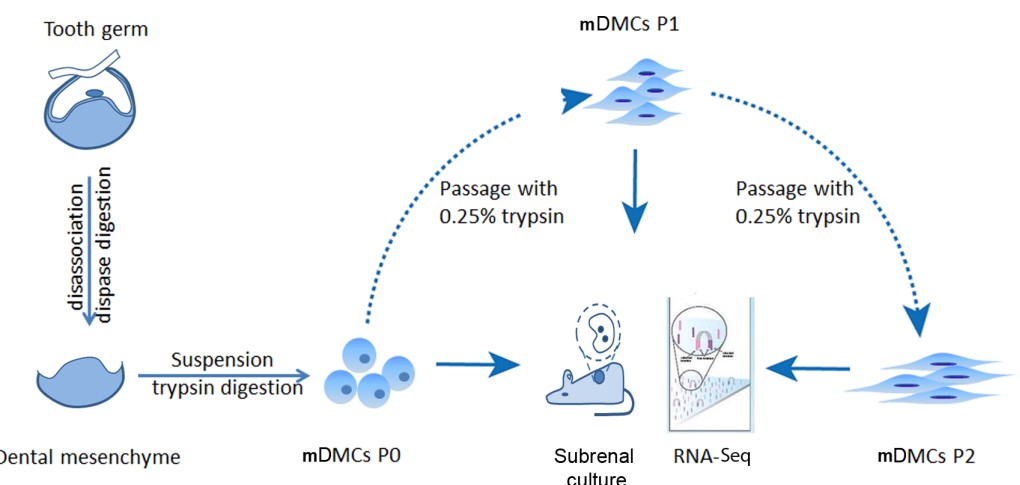

**Figure 1  Experimental design.** Tooth germs from embryonic day 14.5 mice were obtained and digested with dispase to separate the dental mesenchyme from dental epithelium. Freshly isolated dental mesenchymal cells were designated as P0 and cultured *in vitro*. RNA samples from the P0, the first (P1), and second (P2) passages were collected before they were submitted for RNA-seq using an Illumina Hiseq™ 2000.

when recombined with embryonic dental epithelium, cultured mDMCs failed to support tooth development (Fig. 2B). Moreover, the expression of genes that are essential for tooth development were analyzed using qRT-PCR. The expression of *Msx1*, *Pax9*, and *Lhx6* was significantly reduced in P1 and P2 cells compared with P0 cells (Fig. 2C). The expression of *Fgf3* and *Bmp2* in cultured mDMCs was reduced compared with the P0 cells, but the expression of *Bmp4* was not significantly different between the P0 cells and cultured mDMCs (Fig. 2D).

## Overview of the mouse dental mesenchymal cells' transcriptome

To obtain a global view of genes regulating the loss of odontogenic potential, total mRNA of P0, P1, and P2 cells was extracted and sequenced. After data correction, 11,340 transcripts could be matched exactly to known mouse Ensemble transcripts. A total of 9,815 genes were shared among P0, P1 and P2 cells, whereas 563 genes were expressed exclusively in P0 cells (Fig. 3A). P0 cells that were not exposed to *in vitro* culture conditions showed a striking separation from P1 and P2 cells (Fig. 3B; Fig. S1). The transcriptional disparity between freshly isolated and cultured mDMCs is consistent with their phenotypic differences. Differential expression analysis revealed that *in vitro* expansion of mDMCs promoted the selective overexpression of 859 genes, whereas 763 genes were downregulated in P1 cells (Fig. 3C). Comparison of the transcriptomes of P0 and P2 cells revealed that 1,004 genes were upregulated and 948 were downregulated (Fig. 3C). In contrast, 13 genes were upregulated and two genes were downregulated in P1 compared with P2 cells (Fig. 3C). These results suggested that the transcriptome of mDMCs was significantly influenced by *in vitro* culture conditions. In addition, the expression levels of *Msx1*, *Lhx6*, *Pax9*, *Bmp4*, *Fgf10*, *Bmp2*, and *Fgf3* were comparable when analyzed with RNA-seq and qRT-PCR (Fig. 3D).

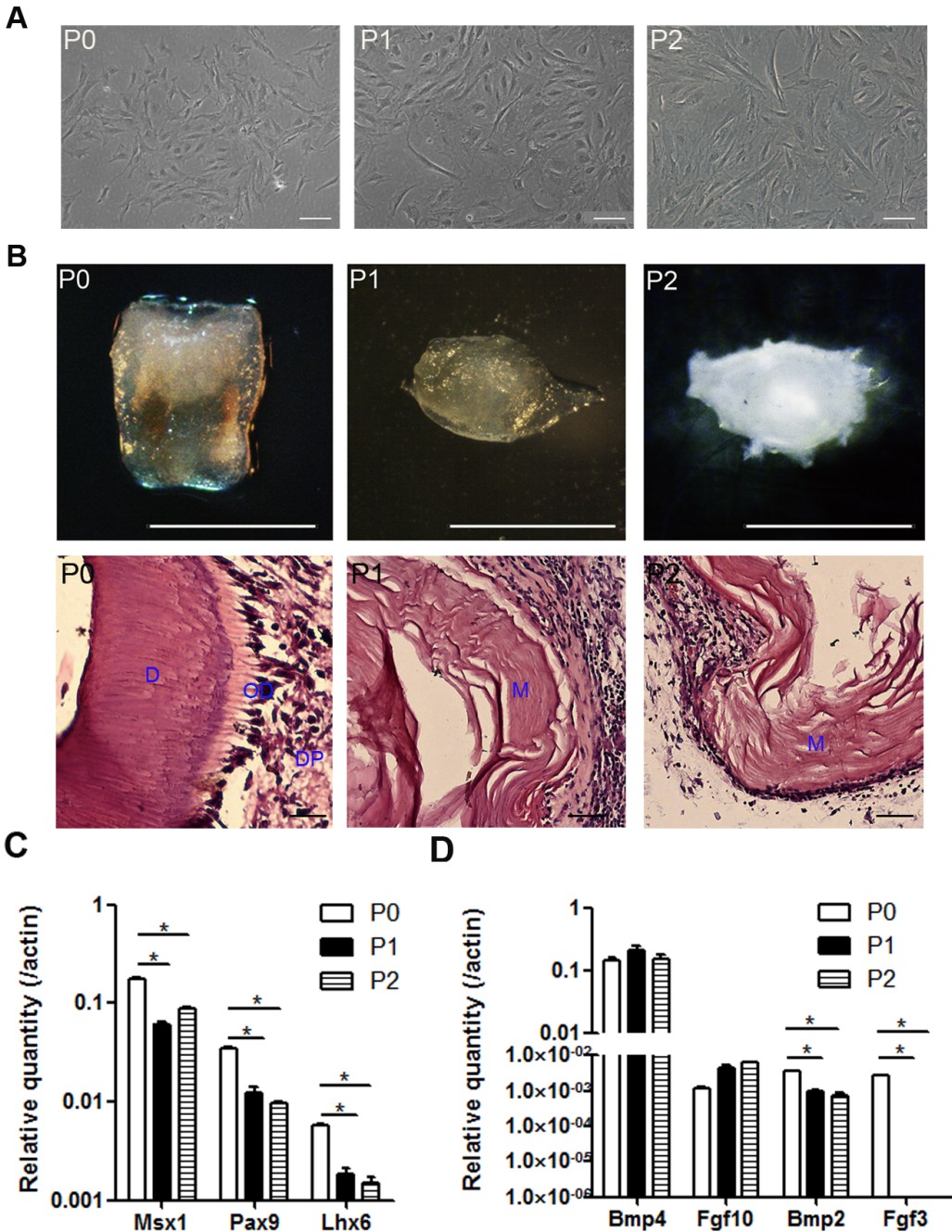

**Figure 2** **The odontogenic potential is impaired in the cultured mDMCs.** (A) mDMCs possessed a fibroblastic-like morphology, and the nuclear-to-cytoplasmic ratio of the P1 and P2 cells was relatively smaller compared with the P0 cells. (B) The P0 cells developed into well-structured teeth when reassociated with dental epithelium, whereas the P1 and P2 cells failed to support tooth development as revealed by the stereoscopic microscopes (upper panel) or HE staining (lower panel; D, dentin; OD, odontoblast; DP, dental pulp; M, matrix. scale bar: upper 1 mm, lower 50 $\mu$m). (C) The mRNA levels of *Msx1*, *Pax9*, and *Lhx6* in mDMCs. (D) The mRNA levels of *Bmp4*, *Fgf10*, *Bmp2*, and *Fgf3* in mDMCs. Data are expressed as the mean $\pm$ standard deviation (SD). *$p < 0.05$.

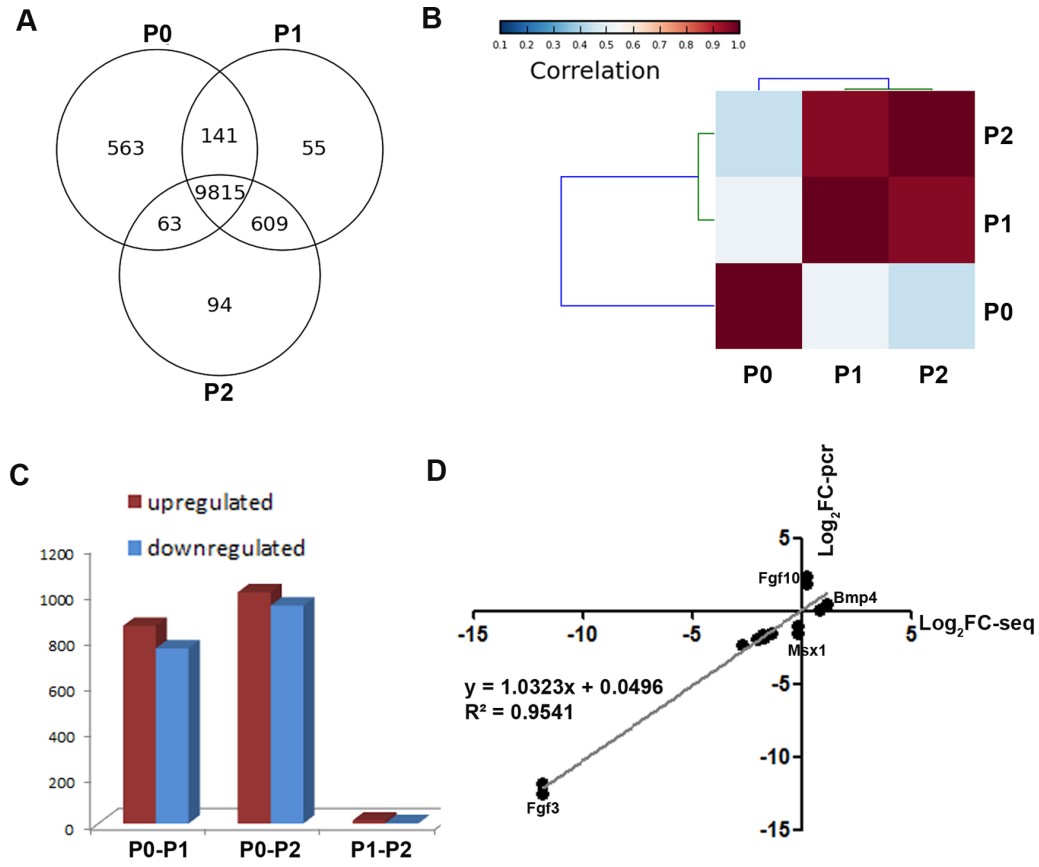

**Figure 3  Comparison of the transcriptomic profiles of freshly isolated and cultured mDMCs.** (A) The Venn diagram shows the expression profiling of P0, P1 and P2 cells, with each section showing the number of genes. (B) Hierarchical clustering analysis showed that the P1 and P2 were highly correlated, whereas the P0 was independent of them. (C) The number of differentially-expressed genes is illustrated. (D) The expression levels of *Msx1*, *Lhx6*, *Pax9*, *Bmp4*, *Fgf10*, *Bmp2*, and *Fgf3* were comparable when analyzed with RNA-seq and qRT-PCR. Data are expressed as the mean ± standard deviation (SD). *$p < 0.05$.

## Gene ontology analysis of differentially expressed genes

Gene ontology (GO) analysis provides an intuitive and effective approach to understand the function of genes in three domains: biological processes, cellular components, and molecular functions. To understand the function of differentially expressed genes, GO analysis was conducted (Fig. S2) and a network diagram was created to illustrate the communication of differentially expressed genes in the enriched clusters of biological processes (Fig. 4). The network is composed of: (a) genes around the node *Dnmt3b*, namely *Dnmt1*, *Ezh2*, *Uhrf1*, and *Rbl1*, which participate in chromatin organization and transcriptional regulation; (b) genes around the node *Runx2*, namely *Bmp2*, *Dlx1*, *Lhx6*, and *Sp7*, involved in cell differentiation and skeletal system development; (c) genes around the node *Fn1*, namely *Robo4*, *Tie1*, *Pecam1* and *Col5a1*, involved in cell motion and cell adhesion; and (d) genes that are not linked to any specific node, whose function is related to apoptosis (*Tradd*, *Tnfrsf11b*) and epithelial morphogenesis (*Grem1*). These various genes

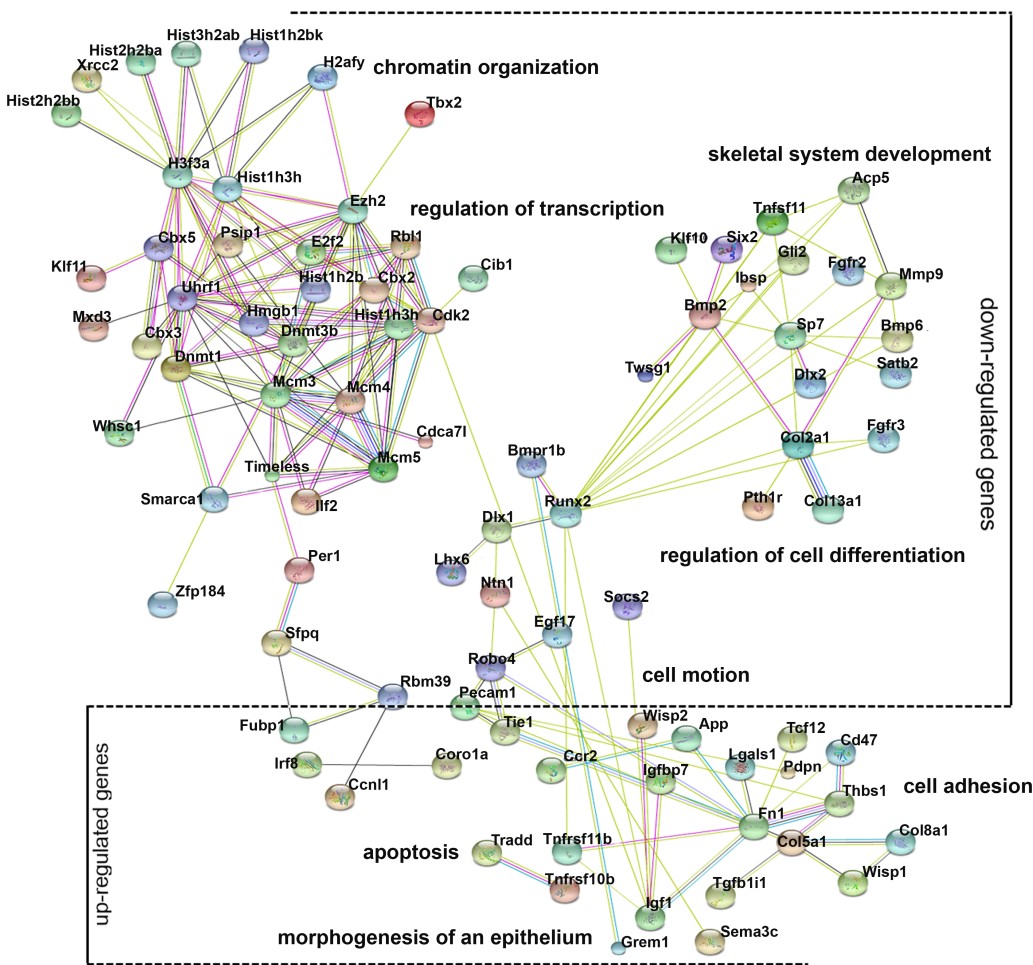

**Figure 4** **Gene ontology (GO) analysis of the differentially expressed genes.** A protein–protein interaction network was constructed. DEGs in enriched GO categories are represented in the network.

and their functions are expected because chromatin modification, cell motion, and cell adhesion are basic cellular functions involving the development of multiple organs.

## Pathway analysis of the differentially expressed genes

Multiple pathways including fibroblast growth factor (FGF), bone morphogenetic protein (BMP), hedgehog (SHH), and Wnt signaling play critical roles in odontogenesis (*Liu et al., 2013*; *Thesleff, Vaahtokari & Partanen, 1995*); thus, the Kyoto Encyclopedia of Genes and Genomes (KEGG) pathway analysis was used to determine the pathways involving the differentially expressed genes (Fig. 5A). Not surprisingly, genes showing a threefold or greater change in expression were related to the pathways regulating basic cellular activities (extracellular matrix-receptor interaction [ko04512], focal adhesion [ko04510], DNA replication [ko03030], and p53 signaling pathway [ko04115]). Notably, essential pathways regulating tooth development were also enriched, including the mitogen-activated protein kinase (MAPK) signaling pathway [ko04010], the transforming growth factor beta (TGFβ) signaling pathway [ko04350], and the Hedgehog signaling pathway [ko04350]. However,

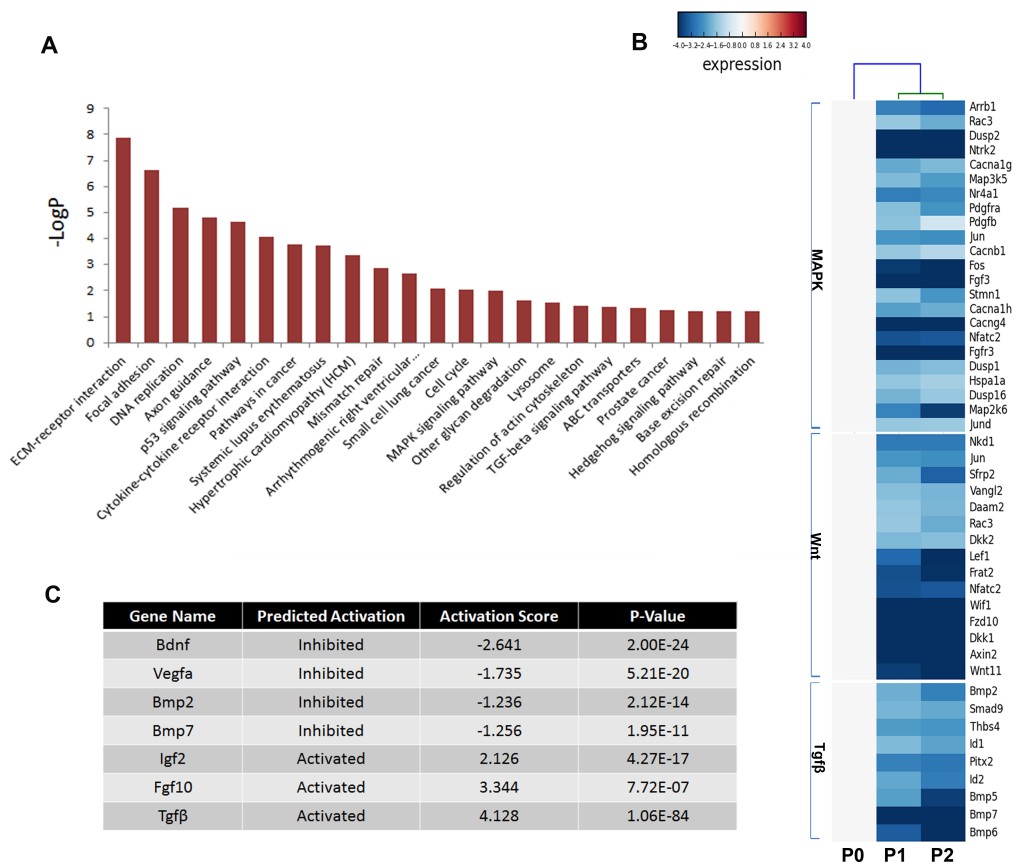

**Figure 5   Disturbed pathways and the upstream regulators.** (A) Kyoto Encyclopedia of Genes and Genomes (KEGG) pathways involving the differentially expressed genes are listed. (B) A heat map representing the genes within the MAPK, Wnt, and TGFβ pathways. (C) The upstream regulators predicted by Ingenuity pathway analysis.

numerous members of these signaling pathways were reduced in cultured mDMCs (Fig. 5B), and it was difficult to identify the critical regulators that are responsible for the loss of odontogenic potential in cultured mDMCs.

An upstream analysis predicts which upstream regulators were most likely to be involved based on the current levels of gene expression detected by RNA-seq analysis. Numerous upstream regulators were related to the loss of odontogenic potential in cultured mDMCs. Specifically, the activities of *Bdnf*, *Vegf α*, *Bmp2*, and *Bmp7* were predicted to be significantly inhibited, while *Igf2*, *Fgf10*, and *Tgfb1* were more active in cultured mDMCs (Fig. 5C). Although *Bdnf* was slightly increased in cultured mDMCs, its receptor *Ntrk2* was significantly decreased (Fig. 5B). The inhibition of *Bdnf* led to the downregulation of genes that promote odontogenesis, such as *Egr1*, *Egr2*, and *Fos* (File S1). Downregulation of *Bmp2* and *Bmp7* probably contributed to the loss of odontogenic potential via the inhibition of *Runx2*, *Sp7*, and *Msx2* (File S1).

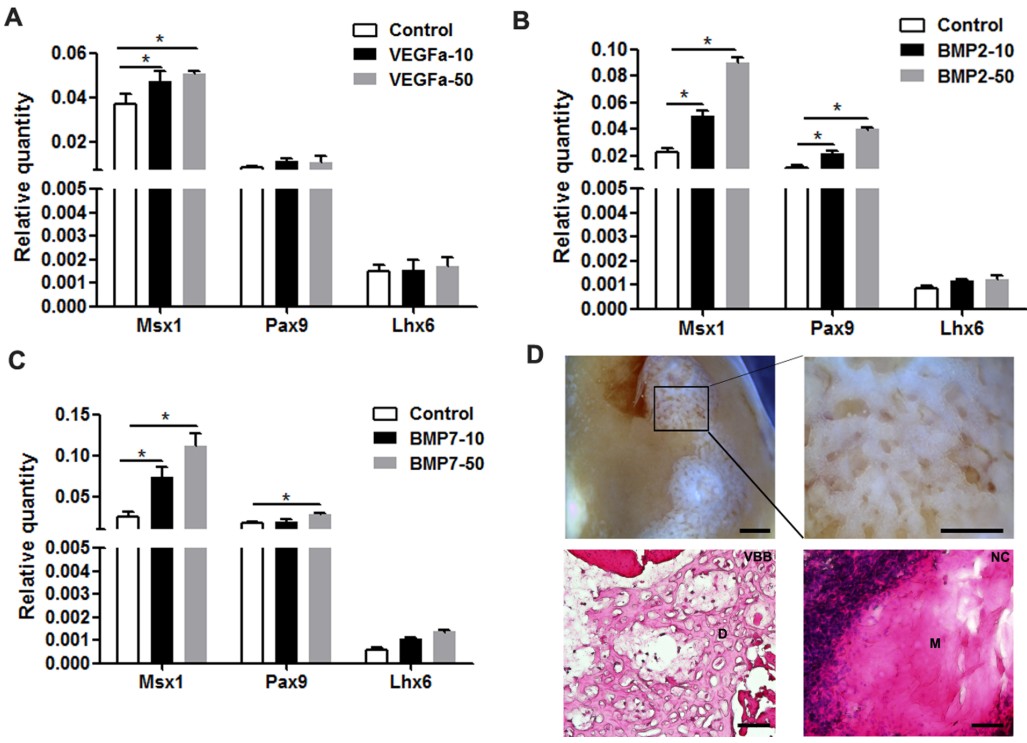

**Figure 6** **VEGFα, BMP2, and BMP7 play roles in the maintenance of odontogenic potential.** The mRNA levels of dental mesenchyme-specific genes in mDMCs cultured in the medium supplemented with 10 ng/ml or 50 ng/ml VEGFα (A), BMP2 (B), and BMP7 (C). (D) Stereoscopic images (upper) and histological images (lower) of the dentin-like structures or the amorphous matrix. Dentin-like structures derived from recombinants with mDMCs cultured in the medium supplemented with VEGFα, BMP2, and BMP7 (VBB). D, dentin-like structures; M, amorphous matrix; NC, negative control. Data are expressed as the mean ± standard deviation (SD). *$p < 0.05$. Scale bar: upper left 1 mm, upper right 500 μm, lower 50 μm.

## VEGFα, BMP2, and BMP7 restore the expression of downregulated genes

Given the pivotal roles of VEGFα, BMP2, and BMP7 in tooth development, we selected these proteins to increase the expression of odontogenic genes. Indeed, tooth mesenchyme-specific genes were upregulated including Msx1 and Pax9 in mDMCs treated with Bmp2 and Bmp7 (Figs. 6B and 6C). VEGFα slightly increased the expression of Msx1 and Pax9, but not that of Lhx6 (Fig. 6A). In parallel, VEGFα, BMP2, and BMP7 were adopted to maintain the odontogenic potential of mDMCs. Although well-structured tooth was not observed, dentin-like structures were present (Fig. 6D). Together, VEGFα, BMP2, and BMP7 promoted the expression of dental mesenchyme-specific genes and the formation of dentin-like structures.

## DISCUSSION

Numerous signaling factors in dental epithelium and/or mesenchyme interact with each other to regulate odontogenesis and induce the expression of dental mesenchyme-specific transcription factors during tooth development (*Thesleff, Vaahtokari & Partanen, 1995*). In

the present study, we found that cultured mDMCs lost the odontogenic potential and failed to initiate tooth formation. Consistently, the dental mesenchyme-specific transcription factors (Msx1, Pax9, and Lhx6) were decreased in cultured mDMCs, and the mutation of these transcription factors was shown to cause tooth agenesis disorders (*Peters et al., 1998*; *Satokata & Maas, 1994*; *Zhao et al., 2013*). The loss of odontogenic potential and decreased expression of dental mesenchyme-specific genes may be due to the absence of inductive signals from the dental epithelium or mesenchyme. Since Bmp and Fgf signaling pathways mediate the odontogenic inductive signals of dental mesenchyme (*Liu et al., 2013*; *Thesleff, Vaahtokari & Partanen, 1995*), the expression of representative components within these two signaling families was analyzed in mDMCs. The expression of Bmp4 shifts from dental epithelium to mesenchyme, which is concomitant with the shift of instructive potential during tooth development (*Vainio et al., 1993*), but the expression of *Bmp4* in the present study was not significantly changed when mDMCs were cultured *in vitro*. This suggests that *Bmp4* in cultured mDMCs is not sufficient to substitute for the odontogenic potential and that other factors besides *Bmp4* are responsible for the inductive potential. Similarly, although *Fgf10* stimulates cell proliferation in the dental epithelium during tooth morphogenesis (*Kettunen et al., 2000*), the mRNA level of *Fgf10* was comparable between freshly-isolated and cultured mDMCs. In contrast, the expression of *Fgf3* and *Bmp2* in cultured mDMCs was significantly reduced. *Fgf3* regulates the number, position, and interrelation of cusps in molar teeth; mutation of the *Fgf3* gene leads to dosage-dependent morphological changes in teeth of both mice and human patients (*Charles et al., 2009*). *Bmp2* is essential for the differentiation of ameloblasts and odontoblasts (*Guo et al., 2015*; *Yang et al., 2012*). Thus, the downregulation of these genes may account for the loss of odontogenic potential in cultured mDMCs. However, *Bmp2* was expressed at a relatively low level in E14.5 mDMCs, and *Fgf3* knockout mice do not exhibit any overt defects in teeth (*Mansour, Goddard & Capecchi, 1993*), raising the possibility that the loss of odontogenic potential in cultured mDMCs may also involve other genes.

Transcriptome analysis, as conducted in this work, has provided insight into the molecular mechanism underlying the loss of odontogenic potential. The transcriptomic profiles reflected the potential of mDMCs, and the mDMCs without odontogenic potential were strikingly separated from those with odontogenic potential. Moreover, the activities of *Bdnf*, *Vegf α*, *Bmp2*, and *Bmp7* were predicted to be inhibited. These growth factors were reported to be involved in tooth development. *Bdnf* and its receptor *Ntrk2* are involved in the MAPK pathway and play roles in epithelial-mesenchymal interactions in early tooth morphogenetic events (*Nosrat et al., 1997*). *Vegfα* is an important factor that induces angiogenesis and related with the MAPK signaling pathway. It is expressed in inner enamel epithelial cells and the basement membrane (*Aida et al., 2005*; *Miwa et al., 2007*), indicating a potential role in the epithelial-mesenchymal interactions. *Bmp2* promotes the maturation of odontoblasts and *Bmp2* conditional knockout mice display abnormal tooth phenotypes with a hypomineralization enamel layer, delayed odontoblast differentiation, abnormal dentin tubules, and decreased tooth-related gene expression (*Guo et al., 2015*; *Yang et al., 2012*). *Bmp7* is also essential for tooth development and its level in the fetal dental mesenchymal cells is higher compared with that in the adult dental
pulp cells, representing a potential mediator of the inductive odontogenic potential of dental mesenchyme (*Gao et al., 2015*). Consistently, we found that VEGFα, BMP2, and BMP7 increased the expression of dental mesenchyme-specific genes. Even though the supplementation with these growth factors did not restore the odontogenic potential of mDMCs, dentin-like structures formed instead of amorphous matrix, suggesting their potential roles in the maintenance of odontogenic potential. Recently, immortalized fetal dental mesenchymal cell lines were established (*Huang et al., 2015*; *Wu et al., 2015*), and a medium supplemented with upstream signaling molecules such as *Vegf α*, *Bmp2*, and *Bmp7* may preserve the biological properties of mDMCs.

Although the effects of *Igf2*, *Fgf10*, and *Tgfb1* were predicted to be enhanced in cultured mDMCs, they are well-known mesenchymal signals that mediate epithelial-mesenchymal interactions during tooth development (*Matsumoto et al., 2011*; *Nakao et al., 2013*; *Vaahtokari, Vainio & Thesleff, 1991*). Interestingly, the level of *Igf2* in dental mesenchyme decreased over time during development because of the hypermethylation of CpG islands (*Khan et al., 2012*); this suggests that the inhibition of *Igf2* activity is required for normal tooth development. However, further studies are still needed to investigate whether the overactivation of these factors leads to abnormal phenotype of teeth, and whether inhibitors of these factors play a role in the maintenance of odontogenic potential.

In the present study, supplementation with *Vegf*, *Bmp2*, and *Bmp7* did not completely restored the odontogenic potential of mDMCs. However, the GO analysis also provided a clue for the establishment of culture system for mDMCs. GO analysis showed that multiple biological processes involving the development of multiple organs were dysregulated when mDMCs were cultured *in vitro*. Specifically, Chromatin modifier enzymes play essential roles in the establishment of transcriptional programs accompanying cell differentiation during tooth development (*Brook, 2009*). Mesenchymal cell migration and compaction are required to induce the expression of critical odontogenic genes and the differentiation of odontoblasts (*Hu, Parker & Wright, 2015*; *Mammoto et al., 2011*). The dysregulation of cell motion also contributed to the compromised potential in cultured mDMCs. Since both the mandible and teeth are mineralized tissue derived from the neural crest (*Chai et al., 2000*), multiple genes acting in skeletal system development have been identified as affecting the odontogenic signaling cascades (*Komori, 2006*; *Nakashima et al., 2002*). Thus, growth factors or cytokines that related with transcriptional regulation, cell differentiation or mineralization, and cell motion may facilitate the maintenance of odontogenic potential.

In conclusion, we assessed the odontogenic potential of *in vitro*-expanded mDMCs and conducted an analysis of their transcriptome profiles. The results revealed that the transition to an *in vitro* setting induced the loss of odontogenic potential and a significant modulation of the overall transcriptome. Genes encoding a wide array of components of MAPK, TGF-β/BMP, and Wnt pathways were significantly downregulated in cultured cells. Our results also indicated that inhibited activities of *Vegf α*, *Bmp2*, and *Bmp7* contributed to the loss of odontogenic potential and supplementation with these growth factors played roles in the maintenance of odontogenic potential in mDMCs.

## ACKNOWLEDGEMENTS

We thank Dr. Andrew Paul Hutchins and Dr. Xiaoshan Wang for their suggestions on data analysis. We thank Prof. Xiaodong Su and Prof. Faming Chen for their suggestions on the revision of the manuscript.

### Funding

This work was supported by the Peking University's 985 Grants, Open Project of Key Laboratory of Regenerative Biology, Chinese Academy of Sciences (KLRB201401), and National Natural Science Foundation of China (No. 813716977; 81570944). The funders had no role in study design, data collection and analysis, decision to publish, or preparation of the manuscript.

### Grant Disclosures

The following grant information was disclosed by the authors:
Peking University's: 985.
Open Project of Key Laboratory of Regenerative Biology, Chinese Academy of Sciences: KLRB201401.
National Natural Science Foundation of China: 813716977, 81570944.

### Competing Interests

The authors declare there are no competing interests.

### Author Contributions

- Yunfei Zheng conceived and designed the experiments, performed the experiments, analyzed the data, wrote the paper, prepared figures and/or tables.
- Lingfei Jia performed the experiments, analyzed the data, prepared figures and/or tables, reviewed drafts of the paper.
- Pengfei Liu, Dandan Yang and Shubin Chen performed the experiments, reviewed drafts of the paper.
- Waner Hu analyzed the data, prepared figures and/or tables, reviewed drafts of the paper.
- Yuming Zhao and Jinglei Cai contributed reagents/materials/analysis tools, reviewed drafts of the paper.
- Duanqing Pei, Lihong Ge and Shicheng Wei conceived and designed the experiments, reviewed drafts of the paper.

### Animal Ethics

The following information was supplied relating to ethical approvals (i.e., approving body and any reference numbers):

All procedures involving animals described in the present study were reviewed and approved by the Animal Care and Use Committee of Guangzhou Institutes of Health and the Peking University in China (permit Number: CMU-B20100106).

## Data Availability

GEO accession number: GSE65164.

## Supplemental Information

Supplemental information for this article can be found online at http://dx.doi.org/10.7717/peerj.1684#supplemental-information.

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
