# Peer review of "Insight into the maintenance of odontogenic potential in mouse dental mesenchymal cells based on transcriptomic analysis"

_PeerJ, doi:10.7717/peerj.1684_

## Round 0.1 · original submission · Major Revisions

As you will see, we have secured 3 reviews. Two of the reviewers believe it is a useful contribution to the field.

Major points to fix:
1. Figure 4 and 9 are unreadable. For figure 4 rework it with a larger font (one reviewer suggested removing it altogether however reworking so that it is readable should be sufficient.

Figure 9 (wagon wheel plots), should be reworked or removed. As you are not actually showing a network but rather the sets of genes connected to those 3 targets, it may be much simpler to summarise this in a supplementary table.

2. Two of the reviewers specifically ask for a matching control image for Fig 6D.

3. Refer to additional literature, and provide further justification for the genes chosen for qRTPCR.

4. Add a caveat (or rebut) acknowledging the possibility that culture condition itself excluded the cells which have odontogenic potential.

Reviewer 1 ·

Basic reporting

This study identifies the genes that maintain the odontogenic potential of mouse dental mesenchymal cells in culture and attempts to restore that function in cultured cells.

Minor issues:
Line 172 – The reference to figures should be Fig, 2C, 2D instead.

The GO analysis did not seem to have any added value to the study. Fig 4 is too complicated and the individual genes too small on the figure to be deciphered. In addition, the results paragraph on “Gene ontology analysis of differentially expressed gene” (line 190-203) is mostly speculative and not discussed further. I think the paper will benefit if this section is left out instead.

Experimental design

The experimental principals are sound. The RNAseq was well validated by the PCR. The pathways analysis and upstream analysis showed possible candidate genes that were investigated with experiments.

Minor issues:
Please explain why you picked the genes Msx1, Pax9, Lhx6, Fgf3, Bmp2 and Bmp4 to run RTPCR, they are important tooth development genes but there are no references to discuss why those genes were picked.

Please include H&E micrograph in Fig 6 like you did in Fig 2 so that we can compare the rescued phenotype.

Validity of the findings

The findings are valid and the multiple approaches with respect to transcriptome analysis add strength to the study. Great use of the pathway and upstream analysis to identify candidate genes for further study.
It will be strengthened with the addition of the micrograph mentioned above.

Additional comments

This is a good study that attempts to establish a protocol that will allow the growing of new teeth in the lab for replacement in the clinic. Although they did not completely succeed, it is a step towards creating a source of cells that will one day allow the replacement of lost teeth with real teeth in the future.

·

Basic reporting

No Comments.

Experimental design

1. The experimental protocol lacks to examine the effect of extracellular matrix. Please access MATRIXOME database (http://dbarchive.biosciencedbc.jp/archive/matrixome/bm/home.html) and click “Bodymap (Tissue-based)”, and then, “Molar (frontal section)” to find suitable matrix to maintain their odontogenic potential.
2. A possibility that culture condition itself excluded the cells which have odontogenic potential, may not be ruled out. Please find the suitable culture conditions instead of adding several growth factors afterwards.
3. Microarray analyses of tooth-like structure of P0, P1, and P2 (Figure 2B) which may include the information of epithelial-mesenchymal interaction are required.

Validity of the findings

The results of this study are predictable since authors did not find novel genes required for odontogenesis.

Additional comments

This paper describes the roles of Vegfa, Bmp2, and Bmp7 on maintaining odontogenic potential in mDMCs using transcriptomic approach. Although the hypothesis of this study is interesting, it seems too preliminary to induce any conclusion.

·

Basic reporting

This is a well written article. It meets the PeerJ policies provided.
There are only minor corrections to be made.
Abstract line 55 delete "of genes".
Line 216 change hard to difficult
Define letters D OD DP M in the figure 2 second lower panel.
Figure 2 D is in the wrong spot and appears to belong to Figure 3, please amend.
Figure 3 D is not explained. Please emend of delete.
Figure 9 can be moved to Supplementary material.

Experimental design

This is a well designed article. It meets the PeerJ policies provided.
I would like to see however a control image for figure 6D in non supplemented cells.

Validity of the findings

It meets the PeerJ requirements provided.

Additional comments

A nice piece of work with a simple approach.

---

## Round 0.2 · Minor Revisions

Dear Dr Wei,
The reviewers are all happy with your revisions, however reviewer 1 has requested a minor edit before it can be accepted. Please make the required change.

Sincerely,
Alistair Forrest

Reviewer 1 ·

Basic reporting

This study identifies the genes that maintain/restore the odontogenic potential of mouse dental mesenchymal cells in culture and attempts to restore that function in cultured cells.
The authors were meticulous in answering the questions that the reviewers had.

Experimental design

The experimental principals are sound. The RNAseq was well validated by the PCR. The pathways analysis and upstream analysis showed possible candidate genes that were investigated with experiments. The additional figures and reworked networks adds to the clarity of the results.

The paragraph and its references from the rebuttal letter on why the GO analysis provides clues to biological processes should be included in the discussion. It will add more weight to the statements made (line 190-203) in the GO results as they currently do not have any other studies backing them and were not explained in the current discussion section.

"Chromatin modifier enzymes play essential roles in the establishment of transcriptional programs accompanying cell differentiation during tooth development (Brook 2009). Mesenchymal cell migration and compaction are required to induce the expression of critical odontogenic genes and the differentiation of odontoblasts (Hu et al. 2015; Mammoto et al. 2011). Thus, the dysregulation of cell motion also contributed to the compromised potential in cultured mDMCs. Since both the mandible and teeth are mineralized tissue derived from the neural crest (Anders & Huber 2010; Chai et al. 2000), multiple genes acting in skeletal system development have been identified as affecting the odontogenic signaling cascades (Komori 2006; Nakashima et al. 2002). In general, when mDMCs were cultured in vitro, the modified epigenetic landscape, reduced mobility, and compromised biomineralization contributed to the loss of odontogenic potential."

Validity of the findings

The findings are valid and the multiple approaches with respect to transcriptome analysis adds strength to the study.

·

Basic reporting

No comments.

Experimental design

The authors responded to my comment and selected to use only one protein, Smoc-1 as a matrix to maintain odontogenic potential of dental mesenchyme cells, but failed. As Professor Yamanaka who developed iPS cells did, the authors may try to select key gene(s) with subtraction method from highly expressed genes in the matrix of dentinogenesis using neutralizing antibody or integrin blocking antibody, in the future study.
Furthermore, to avoid cell death during primary culture, the reviewer recommends to use matrigen (http://www.matrigen.com/) for ideal hardness for mDMCs, in the further study.

Validity of the findings

No comments.

Additional comments

The manuscript has fairly been improved.

·

Basic reporting

Satisfactory

Experimental design

Satisfactory

Validity of the findings

Satisfactory

Additional comments

Suggested changes have been made satisfactorily.

---

## Round 0.3 · accepted · Accept

Dear Dr Wei,
Thank you for this final revision. Congratulations, the paper is now accepted.